# Biotransformation of *p*-xylene into terephthalic acid by engineered *Escherichia coli*

Zi Wei Luo[1] & Sang Yup Lee[1,2,3]

Terephthalic acid (TPA) is an important industrial chemical currently produced by energy intensive and potentially hazardous *p*-xylene (*p*X) oxidation process. Here we report the development of metabolically engineered *Escherichia coli* system for biological transformation of *p*X into TPA. The engineered *E. coli* strain harbours a synthetic TPA pathway optimized through manipulation of expression levels of upstream and downstream modules. The upstream pathway converts *p*X to *p*-toluic acid (*p*TA) and the downstream pathway transforms *p*TA to TPA. In a two-phase partitioning fermentation, the engineered strain converts 8.8 g *p*X into 13.3 g TPA, which corresponds to a conversion yield of 96.7 mol%. These results suggest that the *E. coli* system presented here might be a promising alternative for the large-scale biotechnological production of TPA and lays the foundations for the future development of sustainable approaches for TPA production.

[1] Metabolic and Biomolecular Engineering National Research Laboratory, Department of Chemical and Biomolecular Engineering (BK21 Plus Program), Center for Systems and Synthetic Biotechnology, Institute for the BioCentury, Korea Advanced Institute of Science and Technology (KAIST), 291 Daehak-ro, Yuseong-gu, Daejeon 34141, Republic of Korea. [2] BioProcess Engineering Research Center, KAIST, Daejeon 34141, Republic of Korea. [3] BioInformatics Research Center, KAIST, Daejeon 34141, Republic of Korea. Correspondence and requests for materials should be addressed to S.Y.L. (email: leesy@kaist.ac.kr).

Terephthalic acid (TPA) is a large-volume commodity chemical consumed principally as a monomer precursor in the manufacture of polyethylene terephthalate, which finds a broad range of applications in making beverage containers, fibres, films and other applications[1]. Currently, TPA is chemically synthesized through a series of oxidation reactions from p-xylene (pX), a petroleum-derived raw material. Among various chemical methods for TPA synthesis (Supplementary Table 1), the most notable commercial process, known as the Amoco process, employs a liquid-phase aerobic oxidation scheme[2,3]. Briefly, pX is diluted with glacial acetic acid, and manganese and cobalt salts (for example, acetates) are used as catalysts together with a bromide compound (for example, sodium bromide) added as a promoter[4]. With a typical TPA yield of over 95 mol%, this technology now accounts for nearly all of the TPA produced worldwide.

There are, however, several drawbacks associated with this petroleum-based process, for example, high energy requirement at high temperature (175 ~ 225 °C) and pressure (15 ~ 30 bar), acetic acid loss by solvent burning, requirement of reactor coated with special alloy material, potential hazard to environment and stratospheric ozone by heavy metal catalysts and bromide derivatives[5], and formation of 4-carboxybenzaldehyde (4-CBA) as a byproduct. In comparison with chemical processes, biotransformation processes for chemical production offer several advantages such as reaction operation under milder conditions (for example, ambient temperature and pressure), use of fewer and less toxic chemicals and no need to use expensive and often toxic heavy metal catalysts[6]. Given these evident process advantages, we were motivated to produce TPA through whole-cell biotransformation of pX. Our motivation for bio-based TPA was further enhanced by recent development of several processes for biomass-derived pX production (Supplementary Table 2), such as Gevo's renewable pX from isobutanol[7], fructose to pX through 2,5-dimethylfuran[8] and bio-based ethylene to pX[9]. Thus, if a microbial strain that can biotransform pX to TPA is developed, bio-based renewable production of TPA can be realized.

There have been several attempts to discover and isolate wild-type bacterial strains from nature that are able to convert pX to TPA (Supplementary Table 3). In addition, a recombinant Escherichia coli harbouring a xylene monooxygenase (XMO) was developed to convert pX to 4-carboxybenzyl alcohol (4-CBAL), followed by a chemical process to oxidize 4-CBAL to TPA[10]. However, the conversion efficiencies reported in the above studies were extremely low. Furthermore, in the case of wild-type bacterial isolates, the underlying catalytic mechanisms, that is, genes and encoded enzymes and pathways responsible for each conversion steps, have not yet been elucidated. Thus, it is necessary to first elucidate and validate each reaction steps of the bioconversion process, which can subsequently be implemented for pathway design, strain development and process optimization towards achieving high TPA yield. Here we report the development of a metabolically engineered E. coli strain capable of oxidizing pX into TPA, based on detailed studies on pathway reactions, and reconstruction of synthetic pathway for TPA production. In addition, TPA production was improved through manipulation of enzyme expression levels for increasing the metabolic flux towards TPA biosynthesis, while reducing byproducts formation, and developing a two-phase partitioning fermentation process.

## Results

### Designing a synthetic pX to TPA biotransformation pathway.
We first designed a synthetic metabolic pathway converting pX to TPA (Fig. 1), which can be divided into two parts: upstream pathway responsible for converting pX to p-toluic acid (pTA) and downstream pathway responsible for converting pTA to TPA. The upstream pathway is found in the initial three steps of natural degradation of pX (Supplementary Fig. 1a), which are the sequential oxidation of one of the two methyl groups of pX to its corresponding alcohol (p-tolualcohol, pTALC), aldehyde (p-toluraldehyde, pTALD) and pTA catalysed by XMO, benzyl alcohol dehydrogenase (BADH) and benzaldehyde dehydrogenase (BZDH), respectively. Activities of these enzymes have been identified in a variety of organisms (Supplementary Table 4). In particular, Pseudomonas putida is a well-known natural degrader of various aromatic chemicals including toluene and xylenes (m- and p-xylene)[11]. For example, the TOL plasmid pWW0 in P. putida mt-2 harbours an operon containing the xylMA, xylB and xylC genes encoding XMO, BADH and BZDH, respectively, for initiating the degradation of toluene and xylenes[12,13]. It has been reported that the xylMA-encoded XMO is capable of multi-step oxidation of pX to pTALC and also to pTALD and pTA at less efficiencies[14]. In addition, the xylB-encoded BADH was shown to contribute to the backward conversion from benzaldehyde to benzyl alcohol under the physiological condition[14].

Based on these observations, a synthetic upstream pathway was designed in this study as follows. The P. putida F1 xylMA genes encoding XMO and P. putida KT2440 xylC gene encoding BZDH were chosen for the conversion of pX to pTALD and pTALD to pTA, respectively. The reasons why we did not choose xylB-encoded BADH are as follows: first, it can result in the backward formation of pTALC from pTALD, which compromises the overall flux for TPA production; second, xylMA-encoded XMO can consecutively oxidize pX to pTALD by consuming two molecules of cofactor NADH (together with molecular oxygen), which would allow the whole pX to TPA pathway to be redox neutral (Fig. 1); it is noteworthy that BADH would use NAD$^+$ for the oxidation reaction converting pTALC to pTALD.

The downstream pathway from pTA to TPA can be found from the reactions involved in natural degradation of p-toluene sulfonate and pTA (Supplementary Fig. 1b). The three steps of the downstream pathway are the sequential oxidation of the remaining methyl group of pTA to the corresponding alcohol (4-CBAL), aldehyde (4-CBA) and acid (TPA), catalysed by p-toluene sulfonate monooxygenase (TsaMB), 4-CBAL dehydrogenase (TsaC) and 4-CBA dehydrogenase (TsaD), respectively. These three enzymes are encoded by tsaMB, tsaC and tsaD, respectively, and are present as an operon in the plasmid pTSA of Comamonas testosteroni T-2 (refs 14,15,16). Thus, the downstream pathway converting pTA to TPA was established by employing these three genes. As will be described below, the upstream and downstream pathways were combined to establish a synthetic pathway for the conversion of pX to TPA.

### A synthetic pathway for pX to TPA biotransformation.
All the upstream and downstream pathway genes in the synthetically designed TPA pathway were separately cloned and their expression in E. coli DH5α were confirmed by SDS–polyacrylamide gel electrophoresis (PAGE) (Supplementary Fig. 2), except for XylA subunit of XMO and TsaMB, whose bands could not be seen in the SDS–PAGE analysis. Thus, enzyme functions were examined by whole-cell bioconversion assays using the resting cells expressing each gene or two genes in the operon. Cells expressing XMO (DH5α harbouring the plasmid pTrcMA) converted pX not only to pTALC but also to pTALD (Supplementary Fig. 3a), which agrees with previous finding[14]. In addition, the result suggested production of pTA, but this observation was not reconfirmed. It is interesting to observe fluctuations of pTALC

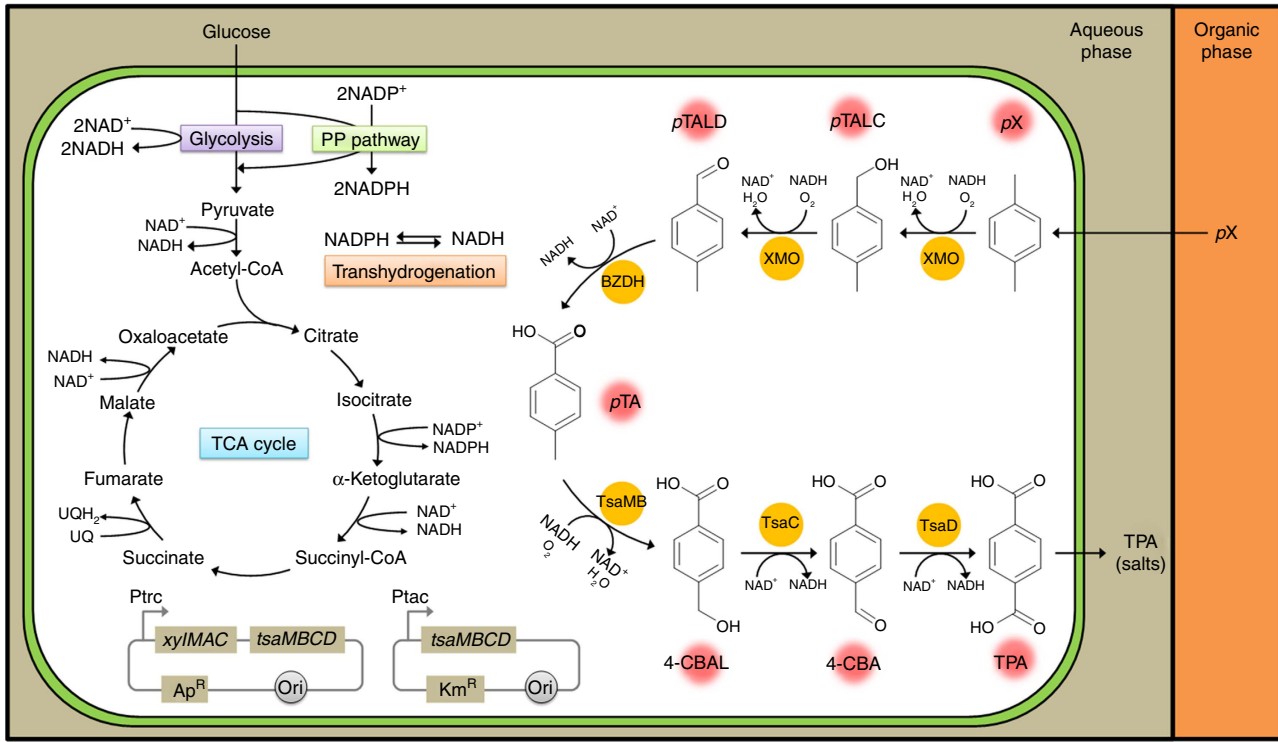

**Figure 1 | Biotransformation pathway from *p*X to TPA and the central carbon metabolism of *E. coli*.** Glucose and *p*X are transported into the cell from the aqueous and organic (OA) phases, respectively. The synthetic pathway biotransforming *p*X to TPA is shown on the right side of the cell. The synthetic pathway was established in *E. coli* through the introduction of two plasmids system shown at the bottom. TPA produced is excreted from cells into the aqueous phase in the form of TPA di-salts at the culture pH of 7.0 employed during the two-phase fermentation. The redox cofactor-related reactions are also indicated. BADH, benzyl alcohol dehydrogenase; BZDH, benzaldehyde dehydrogenase; 4-CBA, 4-carboxybenzaldehyde; 4-CBAL, 4-carboxybenzyl alcohol; $NAD^+$, nicotinamide adenine dinucleotide oxidized form; NADH, nicotinamide adenine dinucleotide reduced form; *p*TA, *p*-toluic acid; *p*TALC, *p*-tolualcohol; *p*TALD, *p*-tolualdehyde; *p*X, *p*-xylene; TPA, terephthalic acid; TsaC, 4-CBAL dehydrogenase; TsaD, 4-CBA dehydrogenase; TsaMB, toluate methylmonooxygenase; XMO, xylene monooxygenase.

and *p*TALD concentrations. This phenomenon seems to be due to the interconversion between *p*TALC and *p*TALD caused by the combined activities of recombinant XMO and endogenous aldehyde-reducing enzymes[17]. Cells expressing BZDH (DH5α harbouring the plasmid pTrcC) converted *p*TALD to *p*TA, but interestingly, *p*TALC was formed much more than *p*TA (Supplementary Fig. 3b). This backward conversion of *p*TALD to *p*TALC revealed the existence of the activities of endogenous aldehyde reduction enzymes in *E. coli*[17]. Cells expressing TsaMB (DH5α harbouring the plasmid pTrcMB) converted *p*TA to 4-CBAL (Supplementary Fig. 3c). TsaC and TsaD were deliberately co-expressed by mimicking the natural *tsaMBCD* operon and the resultant cells harbouring the plasmid pTacCD successfully converted 4-CBAL to TPA (Supplementary Fig. 3d). Having confirmed the correct enzyme functionalities of each catalytic step, the upstream and downstream pathways were separately assembled by constructing plasmids pTrcMAC and pTacMBCD, respectively, and examined for bioconversion of *p*X to *p*TA and *p*TA to TPA, respectively. Both pathways were confirmed to be functional as we designed (Supplementary Fig. 3e,f).

After validating that both the upstream and downstream pathways of *p*X to TPA conversion were functional, the whole combined pathway was assembled into *E. coli* DH5α by transforming both pTrcMAC and pTacMBCD (Fig. 2a; gene expression configuration i, designated as **Pi**). Then, shake flask cultures were performed to examine conversion of *p*X to TPA. As *p*X is nearly insoluble in aqueous medium, highly volatile and toxic to cells[18], a specialized scheme for flask cultivation was designed as follows. A screw-cap flask (250 ml) containing 50 ml of culture medium was connected with a small attached vial into which *p*X was loaded and came into contact with culture medium by evaporation (Supplementary Fig. 4). Cultivation of an engineered DH5α strain harbouring plasmids corresponding to **Pi** in a flask designed as above produced 34.6 mg l$^{-1}$ of TPA (a total of 1.7 mg TPA) from 200 μl *p*X (173.2 mg *p*X) supplied from the attached vial (Fig. 2b,c). On the other hand, *p*TA was accumulated to as high as 129.2 mg l$^{-1}$ (Fig. 2d), which indicates an imbalanced flux through the whole synthetic pathway; obviously, the downstream pathway is a bottleneck.

**Improved TPA production by gene expression balancing.** To de-bottleneck the synthetic *p*X to TPA bioconversion pathway, the expression levels of upstream and downstream pathway modules were varied by employing two different origins of replication (p15A and pBR322) and promoters (*trc* and *tac*). Other promoters such as T5 and T7 promoters were also evaluated, but these experiments were not successful, owing to various problems such as lack of cell growth (for example, T7-driven downstream pathway) or failure to obtain correct positive clones (for example, point mutations in *tsaMB* under T5 promoter). Based on these results, three additional gene expression configurations **Pii** (pTrcMAC-MBCD), **Piii** (pTrcMAC-MBCD and pTacMBCD) and **Piv** (pTacMAC and pTrcMBCD) in addition to **Pi** were designed and constructed (Fig. 2a, right panel). The transcriptional expression levels of all pathway genes in the four gene expression configurations were determined by reverse

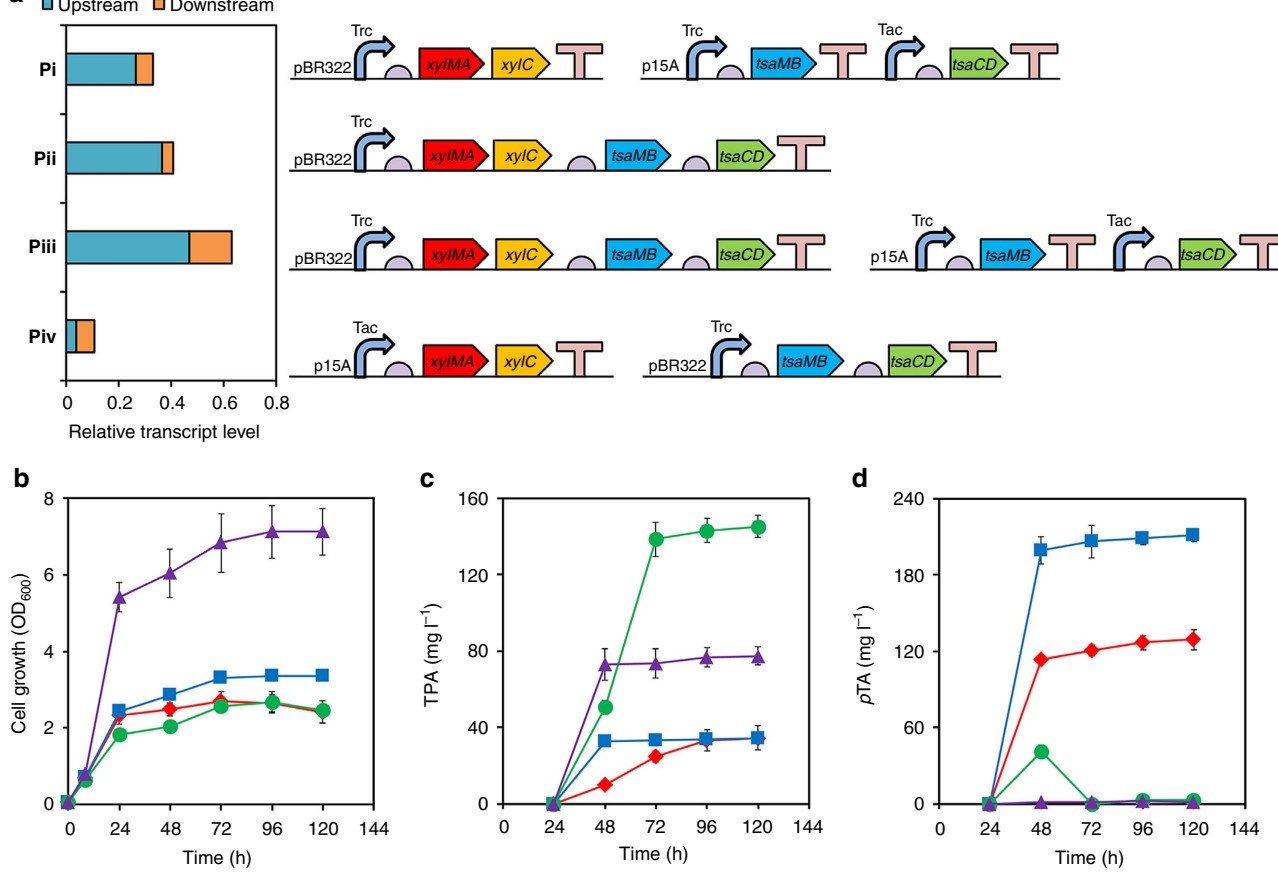

**Figure 2 | Pathway assembly with modular optimization and transformation of *p*X into TPA in shake-flask cultures.** (**a**) Schematic illustration of the constructed plasmids for pathway genes expression (right panel) and the sums of transcript levels of upstream and downstream pathway genes in the four pathway configurations, **Pi**, **Pii**, **Piii** and **Piv** (left panel). Trc and Tac, IPTG-inducible promoters; pBR322 and p15A, replication origins; half-circle box, ribosome-binding site (RBS); T-shape box, terminator. (**b**) Cell growth, (**c**) TPA production and (**d**) byproduct *p*TA formation profiles of *E. coli* DH5α harbouring **Pi** (red diamond), **Pii** (blue rectangle), **Piii** (green circle) and **Piv** (purple triangle) by flask cultures. Values and error bars represent the mean and s.d. of duplicate experiments.

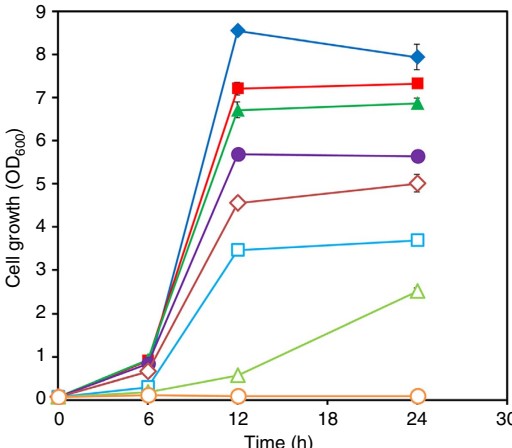

**Figure 3 | Tolerance of *E. coli* to various concentrations of TPA disodium salt.** The symbols represent the concentrations of TPA disodium salt tested as follows: filled diamond, $0 \, \text{g} \, \text{l}^{-1}$; filled rectangle, $5 \, \text{g} \, \text{l}^{-1}$; filled triangle, $10 \, \text{g} \, \text{l}^{-1}$; filled circle, $20 \, \text{g} \, \text{l}^{-1}$; open diamond, $30 \, \text{g} \, \text{l}^{-1}$; open rectangle, $40 \, \text{g} \, \text{l}^{-1}$; open triangle, $50 \, \text{g} \, \text{l}^{-1}$; and open circle, $60 \, \text{g} \, \text{l}^{-1}$. Values and error bars represent the mean and s.d. of repeated experiments.

transcriptase–quantitative PCR (RT–qPCR) (Supplementary Fig. 5) and the sums of transcript abundance for upstream and downstream pathway genes in each gene expression configurations were illustrated (Fig. 2a, left panel). When plasmids corresponding to **Pi**, **Pii** and **Piii** were introduced into *E. coli* DH5α, both cell growth and TPA production were variable (Fig. 2b–d). DH5α harbouring plasmids corresponding to **Pii**, in which the upstream pathway expression level was enhanced compared with **Pi**, produced similar amount of TPA ($34.3 \, \text{mg} \, \text{l}^{-1}$) and more *p*TA ($211.5 \, \text{mg} \, \text{l}^{-1}$). When DH5α harbouring plasmids corresponding to **Piii**, in which the expression levels of both upstream and downstream pathways were elevated beyond **Pii**, was cultured, TPA was produced up to $145.3 \, \text{mg} \, \text{l}^{-1}$ without any accumulation of *p*TA. In DH5α harbouring plasmids corresponding to **Piv**, which showed the least expression levels of upstream pathway genes, the TPA titre was only $77.5 \, \text{mg} \, \text{l}^{-1}$, despite the highest cell density reached among the four constructed expression systems. Furthermore, it was found that the ratio of downstream pathway expression level to upstream pathway (in short, pathway expression ratio) was inversely correlated to the formation of *p*TA. For example, no *p*TA was produced in DH5α harbouring plasmids corresponding to **Piv**, which had the highest pathway expression ratio of 1.90. DH5α

**Table 1 | Two-phase fermentation performances of the engineered *E. coli* strains under various conditions.**

| Strain | Condition | TPA (g l$^{-1}$) | Byproduct* $p$TA (g l$^{-1}$) | Overall productivity (g l$^{-1}$ h$^{-1}$) | Maximum productivity (g l$^{-1}$ h$^{-1}$) | Yield (mol%) |
|---|---|---|---|---|---|---|
| DH5α/**Piii** | 10[†]; 5[‡]; 10[§]; Manual; 1.4[∥] | 2.743 | 1.688 | 0.043 | 0.226 | 57.1 |
| DH5α/**Piii** | 20[†]; 5[‡]; 10[§]; Manual; 1.4[∥] | 1.240 | 2.185 | 0.020 | 0.113 | 31.7 |
| DH5α/**Piii** | 10[†]; 30[‡]; 40[§]; DO-stat; 1.4[∥] | 6.662 | 0.184 | 0.278 | 0.748 | 96.7 |
| DH5α/**Piii** | 20[†]; 30[‡]; 40[§]; DO-stat; 1.4[∥] | 5.844 | 6.007 | 0.126 | 0.861 | 44.4 |
| DH5α/**Piii** | 20[†]; 30[‡]; 40[§]; DO-stat; 0.4[∥] | 6.905 | 1.902 | 0.150 | 0.546 | 74.8 |

OA, oleyl alcohol; $p$TA, $p$-toluic acid; $p$X, $p$-xylene; TPA, terephthalic acid.
*Only $p$TA was detected as the byproduct without other pathway intermediates.
†The amount of $p$X (g) that was fed into the fermenter.
‡The $OD_{600}$ value at which point cell culture was induced.
§The $OD_{600}$ value at which point $p$X feeding was started.
∥The feeding rate of the OA containing dissolved $p$X (ml min$^{-1}$).

harbouring plasmids corresponding to **Piii** had the pathway expression ratio of 0.34, leading to $p$TA detection during the culture but ultimately no $p$TA remained. However, there was significant $p$TA formation in DH5α harbouring plasmids corresponding to **Pi** and **Pii** because of the relatively lower pathway expression ratios (0.25 and 0.12, respectively). Overall, DH5α harbouring plasmids corresponding to **Piii** produced TPA to the highest titre of 145.3 mg l$^{-1}$ (a total of 7.3 mg TPA), a 3.3-fold increase compared with that obtained with a strain harbouring **Pi**. In addition, accumulation of the major byproduct $p$TA was eliminated. Using this *E. coli* DH5α strain harbouring plasmids corresponding to **Piii**, fermentation experiments were performed as described below.

**TPA toxicity to *E. coli*.** Before performing fermentations, the TPA toxicity to *E. coli* was first examined, because it has not been evaluated. As TPA solubility is very low in aqueous solution (for example, solubility in water at 25 °C and 1 atm is 0.0017 g per 100 ml)[19], TPA disodium salt (solubility in water at 25 °C and 1 atm is 14 wt%)[20] was used to prepare various concentrations of TPA salt solutions, which is the main form of TPA (pKa$_1$ of 3.52 and pKa$_2$ of 4.46) present in the culture medium (pH 7.0). Exposing cells to various concentrations of TPA disodium salts up to 10 g l$^{-1}$ (equivalent to TPA concentration of 7.9 g l$^{-1}$) showed only slight reduction in the final optical density (Fig. 3). *E. coli* cells could tolerate up to 40 g l$^{-1}$ of TPA disodium salts (equivalent to TPA concentration of 31.6 g l$^{-1}$). At this TPA disodium salt concentration, the final optical density was a half that of the control without TPA disodium salts (Fig. 3). This relatively high TPA tolerance is particularly important, as *E. coli* can potentially be metabolically engineered to overproduce TPA to a high concentration suitable for industrial applications.

**Two-phase fermentation for TPA production from $p$X.** To overcome the problems (such as volatility, insolubility and toxicity) of feeding $p$X into the fermentor, where strong agitation and aeration are needed to grow cells, a modified organic/aqueous two-phase partitioning bioreactor system[18] was adopted in this study. Oleyl alcohol (OA; 60 wt%, industrial grade) was selected as an immiscible organic carrier phase for two-phase reactor system, owing to its several advantageous characteristics including biocompatibility with a number of microorganisms including *E. coli*, no degradation or utilization by *E. coli*, good phase stability and low cost[21]. Thus, $p$X was dissolved in OA and

allowed to partition into the aqueous phase with a partition coefficient of 524 as determined previously[18]. For example, when 2.1 g $p$X is dissolved in 0.5 l OA, the resulting concentration of $p$X is 8 mg l$^{-1}$ in aqueous phase (1 l)[18]. All two-phase partitioning fermentations in this study were conducted in 2 l MR medium as the aqueous phase and 0.5 l OA as the organic phase in which $p$X was dissolved. Using this two-phase fermentation system, *E. coli* DH5α harbouring plasmids corresponding to **Piii** was examined for TPA production by feeding 10 and 20 g $p$X, respectively. Cells were induced with 1 mM isopropyl β-D-1-thiogalactopyranoside (IPTG) at the $OD_{600}$ of *ca.* 5. OA (0.5 l) containing $p$X was started to be fed into the fermentor at the $OD_{600}$ of *ca.* 10 at a feeding rate of 1.4 ml min$^{-1}$. When 10 g $p$X (in 0.5 l OA) was used, 2.7 g l$^{-1}$ of TPA was produced and 1.7 g l$^{-1}$ of $p$TA was also accumulated (Supplementary Fig. 6a), giving the overall TPA productivity of 0.043 g l$^{-1}$ h$^{-1}$ and conversion yield of 57.1 mol%. When 20 g $p$X (in 0.5 l OA) was used at the same feeding rate, only 1.2 g l$^{-1}$ of TPA was produced along with 2.2 g l$^{-1}$ of $p$TA, giving the TPA productivity of 0.020 g l$^{-1}$ h$^{-1}$ and yield of 31.7 mol% (Supplementary Fig. 6b); the other performance data are summarized (Table 1). In both cases, significant amounts of $p$X (3.8 and 14.3 g, respectively) were lost by evaporation based on the simple $p$X mass balance calculations: (the total amount of $p$X provided) − (the amount of $p$X converted to TPA and $p$TA) − (residual $p$X in the organic phase).

Next, the two-phase fermentation process was further improved to produce TPA to a higher concentration with higher productivity and conversion yield, while minimizing the formation of byproduct $p$TA. The two-phase fermentation process was divided into two stages: biomass production stage and $p$X biotransformation stage. In this manner, batch culture was performed to grow cells to the $OD_{600}$ of *ca.* 30 and then the nutrient feeding for the DO-stat fed-batch culture was started. At the starting point of nutrient feeding, cells were induced with 1 mM IPTG. For the bioconversion of $p$X to TPA, $p$X was started to be fed at the $OD_{600}$ of *ca.* 40. When 10 g $p$X (in 0.5 l OA) was fed at the same feeding rate as above, 6.7 g l$^{-1}$ of TPA was produced and only 0.2 g l$^{-1}$ of $p$TA was accumulated, giving the overall TPA productivity of 0.278 g l$^{-1}$ h$^{-1}$ and conversion yield of 96.7 mol% (Fig. 4). However, when 20 g $p$X (in 0.5 l OA) was fed at the same feeding rate, 5.8 g l$^{-1}$ of TPA was produced with accumulation of 6.0 g l$^{-1}$ of $p$TA, giving the overall TPA productivity of 0.126 g l$^{-1}$ h$^{-1}$ and conversion yield of 44.4 mol% (Supplementary Fig. 7). These results indicated that improved TPA production can be achieved by increasing the

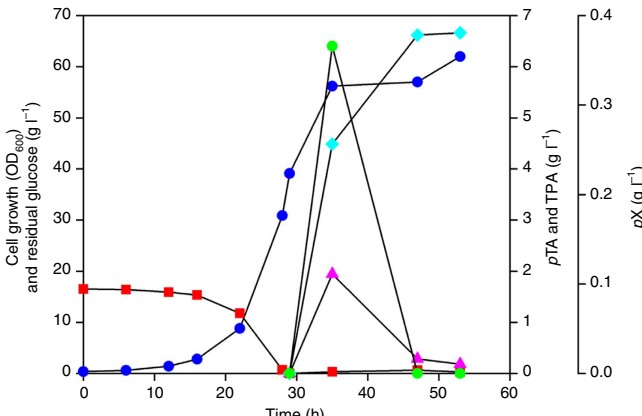

**Figure 4 | Two-phase partitioning DO-stat fed-batch fermentation profile of _E. coli_ DH5α harbouring Piii for TPA production from 10 g of _p_X.** Time profiles of cell growth, residual glucose, _p_X, TPA and _p_TA are shown. Blue circle, cell growth ($OD_{600}$); red rectangle, residual glucose concentration ($gl^{-1}$); pink triangle, _p_TA concentration; light blue diamond, TPA concentration ($gl^{-1}$); green circle, residual _p_X concentration ($gl^{-1}$) in OA.

$OD_{600}$ value before induction and _p_X feeding. It can also be reasoned that a lower _p_X feeding rate appears to be more beneficial for _p_X to TPA conversion. To this end, another two-phase DO-stat fed-batch fermentation was carried out with 20 g _p_X under the same condition, except for using a reduced _p_X feeding rate of $0.4\, ml\, min^{-1}$. This fermentation resulted in the production of $6.9\, gl^{-1}$ of TPA and $1.9\, gl^{-1}$ of _p_TA, giving the TPA productivity of $0.150\, gl^{-1}\, h^{-1}$ and conversion yield of 74.8 mol% (Supplementary Fig. 8). As expected, the TPA titre, yield and productivity were all increased compared with that obtained with _p_X feeding rate of $1.4\, ml\, min^{-1}$. Although the TPA titre was even slightly higher than that obtained with 10 g _p_X at a feeding rate of $1.4\, ml\, min^{-1}$, the conversion yield and overall productivity were both reduced. It can be anticipated that further optimization will allow more efficient production of TPA from _p_X.

## Discussion

In this study, we report a biocatalytic process for the oxidation of _p_X to TPA by developing a metabolically engineered _E. coli_ strain harbouring a synthetically designed pathway. Through designing the synthetic upstream and downstream pathways that convert _p_X to _p_TA and _p_TA to TPA, respectively, followed by optimizing the gene expression levels, TPA could be successfully produced from _p_X. The proof-of-concept two-phase fermentation results described above suggest that it is possible to efficiently produce TPA from _p_X with high conversion yield by employing the engineered _E. coli_ strain developed in this study. As _p_X can be derived from biomass[8–10], TPA can now be produced from biomass rather than fossil resources using the system reported here. Although microbial production of TPA directly from renewable biomass feedstock such as glucose is still not possible, this study provides a hint to future studies on designing a _de novo_ biosynthesis pathway from glucose to TPA. Development of an engineered strain that can produce TPA by direct one-step fermentation from carbohydrates will be an interesting next research topic to study.

## Methods

**Chemicals and bacterial strains**. _p_X, _p_TA and 4-CBAL were purchased from Sigma-Aldrich. _p_TALC, _p_TALD, 4-CBA and TPA were from Acros Organics. OA (60 wt%, industrial grade) was from Tokyo Chemical Industry. _E. coli_ DH5α was

used as host strain for both gene cloning and _p_X to TPA conversion. For plasmid construction and propagation, _E. coli_ DH5α was routinely grown in Luria–Bertani (LB) broth or on LB plates (1.5%, w/v, agar) supplemented with appropriate antibiotics as necessary: $50\,\mu g\, ml^{-1}$ of kanamycin and/or $100\,\mu g\, ml^{-1}$ of ampicillin.

**Plasmid construction.** All the DNA manipulations and cloning for pathway construction were according to the standard protocols[22]. The plasmids and primers used in this study are listed in Supplementary Tables 5 and 6, respectively. To construct pTrcMA, the _xylMA_ genes were amplified from the genome of _P. putida_ F1 using xylMA(pTrc)-f and xylMA(pTrc)-r primers, and cloned into pTrc99A vector amplified using pTrc(xylMA)-f and pTrc(xylMA)-r primers by Gibson assembly[23]. To construct pTrcC, the _xylC_ gene was amplified from the genome of _P. putida_ KT2440 using xylC(pTrc)-f and xylC(pTrc)-r primers, and cloned into pTrc99A vector amplified using pTrc(xylC)-f and pTrc(xylC)-r primers by Gibson assembly. To construct pTrcMAC, the _xylC_ gene was amplified from pTrcC using xylC(pTrcMA)-f and xylC(pTrcMA)-r primers, and assembled with pTrcMA linearized by enzyme digest at _Bam_HI site by Gibson assembly. To construct pTrcMB, the _tsaMB_ genes were amplified from the genomic DNA extract of _C. testosteroni_ T-2 using the primers tsaMB-f and tsaMB-r, and cloned into pTrc99A vector at _Eco_RI and _Bam_HI sites. To construct pTacCD, the _tsaCD_ genes were amplified from the genomic DNA extract of _C. testosteroni_ T-2 using the primers tsaCD-f and tsaCD-r, and cloned into pTac15K vector at _Eco_RI and _Pst_I sites. To construct pTacMBCD, the _trc-tsaMB_-terminator cassette was amplified from pTrcMB using trctsaMB-f and trctsaMB-r primers and assembled with pTacCD linearized by enzyme digest at _Bam_HI site by Gibson assembly. To construct pTrcMAC-MBCD, the RBS-_tsaMB_ cassette amplified from pTrcMB using RBStsaMB-f and RBStsaMB-r primers, and the RBS-_tsaCD_ cassette amplified from pTacCD using RBStsaCD-f and RBStsaCD-r primers were assembled with pTrcMAC linearized by enzyme digest at _Pst_I site by three-fragment Gibson assembly. To construct pTacMAC, the _xylMA-xylC_ cassette was amplified from pTrcMAC using xylMAC(pTac)-f and xylMAC(pTac)-r primers, and assembled with pTac15K vector linearized by enzyme digest at _Eco_RI site by Gibson assembly. To construct pTacMBCD, the RBS-_tsaCD_ cassette was amplified from pTacCD using RBStsaCD(pTrcMB)-f and RBStsaCD(pTrcMB)-r primers, and assembled with pTrcMB linearized by enzyme digest at _Bam_HI site by Gibson assembly. All constructed vectors were confirmed by colony PCR and DNA sequencing.

**SDS–PAGE analysis.** To identify the overexpression of heterologous pathway enzymes XylMA, XylC, TsaMB, TsaC and TsaD (Supplementary Fig. 2), _E. coli_ DH5α cells harbouring the respective recombinant plasmids were collected ($OD_{600}$ $1.0 \times 3\, ml$) from the 10 ml test tube culture with LB medium induced with 1 mM IPTG. Cell pellets were washed with 1 ml of 10 mM $Na_2HPO_4$ buffer (pH 7.2), centrifuged at 16,000 g for 2 min at 4 °C and resuspended in 0.3 ml of the same buffer. Cell lysates of recombinant _E. coli_ were obtained by sonication (High-Intensity Ultrasonic Liquid Processors, Sonics & Material Inc., Newtown, CT). Partially disrupted cells were removed by centrifuging the sonicated samples at 16,000 g for 5 min at 4 °C. The supernatants were used for SDS–PAGE analysis. For identification of the membrane-bound subunit XylM, membrane protein fraction was isolated by further centrifugation at 16,000 g for 30 min at 4 °C, followed by resuspension in 0.5 ml of 10 mM $Na_2HPO_4$ buffer (pH 7.2) containing 0.5% (wt/vol) sarcosyl. After incubation at 37 °C for 30 min, insoluble fraction containing membrane proteins was gained by centrifugation at 16,000 g for 30 min at 4 °C. Membrane proteins were finally prepared by washing the insoluble pellet with 10 mM $Na_2HPO_4$ buffer (pH 7.2) followed by resuspending in 30 μl of TE buffer (pH 8.0).

**RT–qPCR analysis.** The relative abundance of heterologous messenger RNAs was determined using RT–qPCR (Supplementary Fig. 5). Total RNA was extracted from the log-phase culture cells using the Fast HQ RNA Extraction Kit (iNtRON Biotechnology). The preparation of complementary DNA and qPCR were performed in a single step by using the One Step SYBR PrimeScript RT–PCR Kit (TaKaRa, Clontech) and on a LightCycler 96 Instrument (Roche Diagnostics), according to the manufacturer's instructions. The primers used for the genes _xylM_, _xylA_, _xylC_, _tsaM_, _tsaB_, _tsaC_ and _tsaD_ in RT–PCR experiments were listed in Supplementary Table 6. The 16S ribosomal RNA was used as the reference gene for relative quantification.

**Whole-cell activity assay.** The whole-cell activity assay using _E. coli_ resting cells was conducted as follows[13]. Recombinant _E. coli_ cells were incubated in 50 ml of LB medium in the presence of antibiotics as described above. At the $OD_{600}$ of 0.6, cells were induced by the addition of 1 mM IPTG. The incubation was continued for 4–6 h. When the $OD_{600}$ increased to around 2.0, cells were harvested and resuspended in 7 ml of 50 mM potassium phosphate buffer (pH 7.4) containing 1% w/v glucose (Supplementary Fig. 3). Aliquots of 1 ml cell suspension were distributed in 1.5 ml Eppendorf microcentrifuge tubes and horizontally incubated in a shaken incubator at 200 r.p.m. and 30 °C. After 5 min, each respective substrates, _p_X ($100\, mg\, l^{-1}$), _p_TALD ($200\, mg\, l^{-1}$), _p_TA ($200\, mg\, l^{-1}$) and 4-CBAL

($280 \, mg \, l^{-1}$), were added from a stock solution, where $pX$ and $pTALD$ were dissolved in ethanol and $pTA$ and 4-CBAL were dissolved in the assay buffer. The reactions were carried out in the shaking incubator and stopped for sampling at designated time points by placing the tubes on ice. And immediately, sample cells were pelleted by centrifugation at $4 \, ^\circ C$, $16,000 \, g$ for 10 min and then the supernatants were analysed with appropriate dilution.

**TPA toxicity test.** The potentially toxic effect of TPA on microbial cell growth was examined using wild-type *E. coli* strain W3110 in flask culture containing the same medium as described below for TPA production (Fig. 3). Seed cultures were grown in test tubes containing 10 ml of LB medium. Then, 1 ml of seed culture was transferred into a 300 ml baffled flask containing 50 ml of fermentation medium supplemented with TPA disodium salt at various concentrations and cultivated under $37 \, ^\circ C$ in a shaken incubator at 200 r.p.m. The medium was prepared from a TPA disodium salt stock solution of $100 \, g \, l^{-1}$. Control experiments were conducted under the same conditions, except for the TPA disodium salt supplementation. Samples were periodically withdrawn to monitor the cell growth ($OD_{600}$) and the time profiles of cell growth were compared.

**Cultivation.** The screw-cap flasks (250 ml) connected with an attached vial were designed and used for $pX$ to TPA biotransformation in shake-flask cultures (Supplementary Fig. 4). During cultivation, $pX$ was added at indicated time points into the vial and continuously evaporated to contact with the aqueous medium. Once $pX$ was added into the attached vial, the flasks were tightly closed and samples were withdrawn every 24 h. During sampling, the flasks were opened and sampling was quickly performed to minimize $pX$ loss by evaporation. Flask cultures were conducted in MR medium supplemented with $10 \, g \, l^{-1}$ glucose and $10 \, mg \, l^{-1}$ of thiamine HCl. The MR medium (pH 7.0) contained per litre: $6.67 \, g \, KH_2PO_4$, $4 \, g \, (NH_4)_2HPO_4$, $0.8 \, g \, MgSO_4 \cdot 7H_2O$, $0.8 \, g$ citric acid and 5 ml trace metal solution[24]. Glucose, thiamine HCl, MR salt medium, $MgSO_4 \cdot 7H_2O$ and trace metal solution were sterilized separately. Cells were inoculated from glycerol stock into 25 ml test tubes containing 10 ml LB medium and cultivated in a rotary shaker at 200 r.p.m. at $37 \, ^\circ C$. After 12 h, 1 ml of seed culture was transferred into 50 ml MR medium in the designed flask at $37 \, ^\circ C$, 200 r.p.m. When cells grew up to the $OD_{600}$ of around 0.6, the cell culture was induced with 1 mM IPTG and transferred to a rotary shaker at $30 \, ^\circ C$, 200 r.p.m. When necessary, antibiotics were added to the culture medium as described above.

Two-phase partitioning fermentations were carried out in a 6.6 l jar fermentor (Bioflo 3000; New Brunswick Scientific Co., Edison, NJ) containing 1.8 l of MR medium supplemented with $20 \, g \, l^{-1}$ of glucose and $10 \, mg \, l^{-1}$ of thiamine HCl at $30 \, ^\circ C$. The seed culture (200 ml) was prepared as follows. First, glycerol cell stock was used to inoculate a 10 ml test-tube culture with LB medium. After overnight cultivation at $37 \, ^\circ C$ and 200 r.p.m., 4 ml of cell culture was transferred into 500 ml baffled flask containing 200 ml MR medium supplemented with $10 \, g \, l^{-1}$ of glucose and $10 \, mg \, l^{-1}$ of thiamine HCl. Cells were cultured for 12 h and inoculated into the fermentor. The culture pH was controlled at 7.0 using 50% ammonia solution. The dissolved oxygen was controlled at 40% (v/v) by flowing $2 \, l \, min^{-1}$ of air and automatically changing the agitation speed from 200 to 1,000 r.p.m. In the preliminary runs, cells were induced with 1 mM IPTG when the $OD_{600}$ value was *ca.* 5. When the $OD_{600}$ value reached *ca.* 10, OA (0.5 l) containing 10 or 20 g $pX$ were fed into the fermentor via a peristaltic pump at indicated rates. When the glucose concentration of the culture broth decreased below $5 \, g \, l^{-1}$, 50 ml of nutrient feeding solution was manually added to adjust the glucose concentration in the fermentor at *ca.* $20 \, g \, l^{-1}$. The feeding solution contained per litre 700 g glucose, 10 g $MgSO_4 \cdot 7H_2O$, 10 mg thiamine HCl and 5 ml trace metal solution. In the latter runs, batch culture was performed to grow cells to the $OD_{600}$ of *ca.* 30 and then the nutrient feeding for the DO-stat fed-batch culture was started. At the starting point of nutrient feeding, cells were induced with 1 mM IPTG. When the $OD_{600}$ value reached *ca.* 40, OA (0.5 l) containing 10 or 20 g $pX$ were fed into the fermentor via a peristaltic pump at indicated rates. The nutrient feeding solution was the same as above.

**Analytical procedures.** Cell growth was monitored by measuring the optical density at 600 nm ($OD_{600}$) with Ultrospec 3100 spectrophotometer (Amersham Biosciences, Uppsala, Sweden). The concentration of glucose and organic acids (acetate, lactate and formate) were measured by highHPLC (Waters 1515/2414/2707, Waters, Milford, MA). For measurement of TPA and other pathway intermediates, HPLC (1100 Series, Agilent) equipped with a Zorbax SB-Aq column ($4.6 \times 250 \, mm$, Agilent) was used and operated at $30 \, ^\circ C$. The mobile phase consisted of buffer A (25 mM potassium phosphate buffer, pH 2.0) and buffer B (acetonitrile), and was flown at $0.8 \, ml \, min^{-1}$. The following gradient was applied: 0–1 min, 100% A; 1–10 min, a linear gradient of B from 0% to 70%; 10–19 min, 70% B; 19–20 min, a linear gradient of B from 70 to 0%. The injection volume was 5 μl and detection was performed by photodiode array at 240 nm for $pTA$, 4-CBAL, 4-CBA and TPA, and 215 nm for $pTALC$ and $pTALD$. Samples were diluted properly and then filtered through a 0.22 μm syringe filter before HPLC analysis. Standard solutions of $pTA$, 4-CBAL, 4-CBA and TPA were prepared by dissolving each authentic chemical in ammonia and then diluted with distilled water to make up a series of concentrations for calibration curve, whereas $pTALC$

and $pTALD$ were dissolved in acetonitrile. The concentration of residual $pX$ in OA during two-phase cultivations was quantified by gas chromatography (7890, Agilent Technologies) equipped with 80/120 Carbopack B AW packed glass column (Supelco, Bellefonte, PA) and a flame ionization detector (Agilent Technologies).

**Data availability.** The data supporting the findings of this study are available from the authors upon reasonable request.

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

## Acknowledgements

We thank Dr Byoungjin Kim for helpful discussion. We also thank Moon Hee Lee for his help on liquid chromatography and gas chromatography analyses. This work was supported by the Intelligent Synthetic Biology Center through the Global Frontier Project (2011-0031963) of the Ministry of Science, ICT & Future Planning through the National Research Foundation of Korea.

## Author contributions

S.Y.L. generated the idea. Z.W.L. and S.Y.L. designed the project. Z.W.L. performed experiments and analysed the data. Z.W.L. and S.Y.L. wrote the manuscript.

## Additional information

**Competing interests:** Authors declare that they have conflict of interest as the technology described here is patent filed (KR-10-2015-0066686) for potential commercialization.

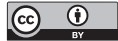

