## [Peer Review File · Nature Communications]

Reviewers' Comments:

Reviewer #1 (Remarks to the Author):

The manuscript by Zuo and Lee reports on the biotransformation of terephthalic acid (TA) from p-xylene by heterologous gene expression in *E. coli*. TA is a globally important, large-volume commodity chemical that is known for its use in PET polymers. Usually, it is chemically produced from fossil-oil derived p-xylene at high temperatures and pressures. For this reason there will be a demand for biotechnological alternatives for PA formation in the future. While recent progress has been achieved in the sustainable synthesis of p-xylene from biomass-derived sources, attempts for the biotransformation of TA from p-xylene have not resulted in satisfying yields, yet.

The authors impressively achieved the production of TA from p-xylene with a yield of 96.9 mol% (13.3 g TA from 8.8 g p-xylene) by expressing seven genes involved in the oxidation of methyl groups to carboxylic acid functionalities. The yield is without doubt a milestone in the field and of great interest for researchers studying sustainable alternatives for petroleum derived chemicals. The authors designed a smart redox-neutral pathway comprising the combined action of oxygenases and dehydrogenases, and provide a number of optimization steps including different arrangement of promoters on expression plasmids or the two-phase fermentation system. In summary, the work provided is not only a proof-of-principle but also provides a promising approach for future large-scale biotechnological TA production.

The manuscript requires some editing, and especially the optimization approaches have been described in too much detail. I recommend that the authors should more focus on the successful approaches and reduce the less successfully designed experiments to a minimum. They should rather summarize the important parameters for optimized TA production.

Other points:

1. L.68: in Ref. 10 the heterologously expressed gene was of course known, so this comment is not really right.
2. L.70: detailed reaction mechanisms have not been studied in this work, and they are also not important for the purpose of this study.
3. Table S4: 'candidates for activities' sounds odd; just candidates is sufficient
4. L.118: have the proteins bands been confirmed by MS? Otherwise the gel is not very informative. The authors should also give the expected molecular masses of the individual gene

products in Fig. S2

5. L.122: I have some problems with Fig. S3: (i) the abbreviations of the plasmids are not yet defined here (they are defined later); (ii) the authors should indicate in the fig. legend which substrate was added in which experiment. (iii) the authors did not show p-xylene consumption in Fig. S3a? (iv) The authors should comment on the up and down in Fig. S3a. (v) In Fig S3b the amount of products formed is much higher than that of the educt consumed, there must be a mistake.

6. L.147-L.180: though I appreciate that the authors tried several gene/promotor arrangements in their expression plasmids, this part needs to be reduced to at least 50% (better even less). The text is not much more as what can be seen in Fig. 2b/ Fig. legends.

7. L.182-187: it is trivial that TA is present as dianion at the pH of culture conditions: reduce this statement to a minimum

8. L.196-L.243: Again this part reporting on the optimization of the fermentation should be shortened to 50%, the authors should rather report on the best conditions and then summarize what parameters were essential.

Reviewer #2 (Remarks to the Author):

Our society is totally dependent in a multitude of ways upon chemicals provided by the chemical industry. However, efforts to achieve sustainable development necessitate departure from polluting and high level energy-consuming processes based on non-renewable feedstocks, through exploitation of environmentally friendly chemistry options. A main plank in sustainable chemistry procedures is biocatalysis. Exploitation of biocatalytic procedures requires that they be economically competitive. Significant hurdles for biocatalytic processes for bulk chemicals are modest yields and susceptibility of the biocatalyst to substrate-intermediate-product toxicity.

In this paper, the authors propose a (ultimately) biocatalytic route from the renewable substrate glucose to terephthalic acid (TPA), the monomer of the commodity polymer PET, though they describe here an intermediate process of conversion of p-xylene to TPA, as a proof-of-principle.

The results presented constitute an elegant biocatalysis process based on a rationally designed metabolic pathway constructed in *E. coli* from the upper xylene degradation pathway of *Pseudomonas putida* F1 and the lower p-toluene sulfonate pathway of *Comamonas testosteroni* T-2. The biocatalytic activity of the construct was assessed in a 2-phase bioconversion system in which the xylene was supplied in the apolar phase. Initial experiments revealed bottlenecks within the designed pathway and several inducible constructs were made to relieve them and optimise product yield. The potential toxicity of TPA caused by high substrate feeding rates was analysed and shown to be unproblematic. In one case, a yield of 97% was recorded, which is

extraordinary and, if reproducible when upscaled, will represent a breakthrough in hydrocarbon bioconversions.

All in all, this is an excellent and important paper documenting the feasibility of bioconversions for bulk chemistry. While not yet documenting a process of conversion of a renewable substrate to TPA, it shows proof-of-principle, since other publications show the potential feasibility of conversion of renewables to p-xylene. In any case, the exploitation of petroleum constituents as feedstocks for chemicals production, despite being unsustainable over long time spans, is, unlike the use of fossil fuels for energy generation, not a sustainability priority, so a scale-up of the process described in this submission, and its implementation as a viable process for large scale production of TPA, may be envisaged.

I congratulate the authors on this important advance and only have one, rather trivial question: they examined the toxicity of TPA applied externally but, in this process, TPA is produced internally. Perhaps they may wish to comment on this, and on TPA release after formation?

Reviewer #3 (Remarks to the Author):

In this manuscript, the authors describe the development of an *E. coli* strain that can convert p-xylene (pX) to terephthalic acid (TPA). The engineered pathway relies on known degradative routes for p-xylene and p-toluene sulfonate. The authors first demonstrate independent validation of the up- and downstream pathways, then demonstrate a two-phase partitioning bioreactor to achieve conversion of pX to TPA. This is an interesting, largely well-written manuscript. The suitability for this journal arises from the nature of the product being targeted – a very high-volume chemical compound – and the potential for achieving completely bio-based production of TPA. The manuscript would be considerably strengthened by addressing the comments below to improve clarity:

1. p. 4, lines 73 and 74 – I don't think it's quite correct to start that there were "detailed studies" on enzymes in this paper. The pathway relies on known pathways and largely confirms previous observations rather than providing any new biochemical insights. The authors are better served by limiting this statement to "pathway reactions" since that is consistent with both the context and the results.
2. p. 5, lines 102-104 – As written, this sentence suggests that BADH does not use NADH and thus would result in a cofactor imbalance. According to Metacyc, the cofactor requirements for BADH are the same as for the second oxidative step of XMO. Thus, I don't see what using XMO provides an advantage with respect to redox balance.

3. p. 6, lines 121-122 and Supp Fig 3a – Is the production of pTA in this figure actually significant? It seems that a small amount may have been detected at 90 min, but without error bars, it's impossible to know if this is "real"? Were there biological replicates that consistently showed pTA production? Any showing accumulation, i.e., detection at more than one time point?
4. p. 9, line 183 – suggest changing "has not been known" to "has not been evaluated" or something similar
5. p. 9, lines 187-194 and Supp Fig 6 – The data do not support the conclusion that there is no effect on growth at TPA-salt concentrations up to 10 g/L. There is a clear difference in max OD between the 0 g/L control and the 5 and 10 g/L samples. It may be true that the growth rates during exponential phase are not different, but there are not enough data points to accurately calculate a growth rate given that there is only one data point between inoculation at time $t=0$ and stationary phase. If the authors want to make claims about growth rate, more data points are needed to ensure that measurements are being made during the exponential phase.
6. p. 15, line 324 – The amount of substrates added need to be provided with more detail than "an appropriate amount." For each experiment, the specific substrate and amount added should be provide. A similar comment is provided below in reference to Supp Fig 3.
7. p. 17, lines 371-373 – For these latter runs, was there also just a single bolus addition of 50-ml nutrient solution? It is noteworthy that Fig 3 shows no accumulation of glucose after the onset of feeding while Supp Fig 7 clearly shows a spike. This suggests a difference in either feeding strategy or physiology that should be explained.
8. Supp Fig 2 – Additional descriptions are required for the arrows, i.e., specify the proteins that are being pointed to, along with the expected molecular weights.
9. Supp Fig 3 – More details are needed here about the experimental setup, in particular with respect to the substrates added in each panel, both the identity and amount. Panels 3a and 3e show productivity originating from pX, but pX titer is not shown anywhere. I understand that the solubility is quite low and thus it may have been unmeasurable, but if this is the case, it should be noted in the figure legend and/or the methods. Panel 3a also shows spikes in metabolites, which suggests some sort of feeding regimen. It would also be very helpful to use the same abbreviations (in parentheses) in the figure legend as are used in the main text.
10. Supp Table 5 is incorrectly listed as Supp Figure 5. Also, the units for pX added (superscript note 'b') need to be provided.

Responses to the Comments

Editor's Comments

Dear Dr Lee,

Your manuscript entitled "Biotransformation of p-xylene into terephthalic acid by engineered Escherichia coli" has now been seen by 3 referees. You will see from their comments below that while they find your work of interest, some important points are raised. We are interested in the possibility of publishing your study in Nature Communications, but would like to consider your response to these concerns in the form of a revised manuscript before we make a final decision on publication. We therefore invite you to revise and resubmit your manuscript, taking into account the points raised. Please highlight all changes in the manuscript text file.

Best regards,

Chuanfu An, Ph.D. Associate Editor, Nature Communications

[Response]: Thank you very much. Although the reviewers' comments were rather minor, they were invaluable in making our paper much clearer and improved. We would like to thank you and the reviewers for the comments.

Reviewers' comments:

Reviewer #1 (Remarks to the Author):

The manuscript by Luo and Lee reports on the biotransformation of terephthalic acid (TA) from p-xylene by heterologous gene expression in E. coli. TA is a globally important, large-volume commodity chemical that is known for its use in PET polymers. Usually, it is chemically produced from fossil-oil derived p-xylene at high temperatures and pressures. For this reason there will be a demand for biotechnological alternatives for PA formation in the future. While recent progress has been achieved in the sustainable synthesis of p-xylene from biomass-derived sources, attempts for the biotransformation of TA from p-xylene have not resulted in satisfying yields, yet.

The authors impressively achieved the production of TA from p-xylene with a yield of 96.9 mol% (13.3 g TA from 8.8 g p-xylene) by expressing seven genes involved in the oxidation

of methyl groups to carboxylic acid functionalities. The yield is without doubt a milestone in the field and of great interest for researchers studying sustainable alternatives for petroleum derived chemicals. The authors designed a smart redox-neutral pathway comprising the combined action of oxygenases and dehydrogenases, and provide a number of optimization steps including different arrangement of promoters on expression plasmids or the two-phase fermentation system. In summary, the work provided is not only a proof-of-principle but also provides a promising approach for future large-scale biotechnological TA production.

The manuscript requires some editing, and especially the optimization approaches have been described in too much detail. I recommend that the authors should more focus on the successful approaches and reduce the less successfully designed experiments to a minimum. They should rather summarize the important parameters for optimized TA production.

[Response] Thank you so very much for appreciating the importance of our work. Regarding your suggestion to reduce the optimization approaches, the other two reviewers are fine with the current level of details. Thus, we think that this level of detail is in fact needed for the other researchers to follow what we did for strain development and bioconversion process development. If editor thinks that we should reduce the details, we will do so. Thank you very much again.

Other points:

1. L.68: in Ref. 10 the heterologously expressed gene was of course known, so this comment is not really right.

[Response]: Thank you. We originally wanted to emphasize the limitation of using wild-type bacterial isolates from nature for TPA production from *pX*. To make this point clear, the sentence is now changed to “Also, in the case of wild-type bacterial isolates, the underlying catalytic mechanisms...”.

2. L.70: detailed reaction mechanisms have not been studied in this work, and they are also not important for the purpose of this study.

[Response]: Thank you. As the reviewer suggested, we rephrased the sentence to “Thus, it is necessary to first elucidate and validate each reaction steps of the bioconversion process...”.

3. Table S4: ‘candidates for activities’ sounds odd; just candidates is sufficient.

[Response]: Thank you. We removed “activities” as suggested.

4. L.118: have the proteins bands been confirmed by MS? Otherwise the gel is not very informative. The authors should also give the expected molecular masses of the individual gene products in Fig. S2

[Response]: Thank you for the comment. It is a great idea to show the expected molecular masses. As suggested, we provided the information on molecular weights of the individual gene products in Supplementary Figure 2 in the revised manuscript.

5. L.122: I have some problems with Fig. S3: (i) the abbreviations of the plasmids are not yet defined here (they are defined later); (ii) the authors should indicate in the fig. legend which substrate was added in which experiment. (iii) the authors did not show *p*-xylene consumption in Fig. S3a? (iv) The authors should comment on the up and down in Fig. S3a. (v) In Fig S3b the amount of products formed is much higher than that of the educt consumed, there must be a mistake.

[Response]: Thank you for these great comments. Our responses are as follows. (i) We defined the abbreviations of the plasmids relevant to Supplementary Fig. 3 in the revised manuscript. (ii) We indicated the substrate information in the revised manuscript. (iii) The reason for which *p*X consumption profile was not shown is that the whole-cell activity assay was conducted in aqueous solution in which *p*X could not be dissolved due to its insolubility in water. Thus, it is not possible to accurately monitor *p*X consumption (this point was already understood by Reviewer 3). (iv) Indeed, it is interesting to see the up and down in the profile of *p*TALC and *p*TALD formations in the Supplementary Fig.3a. We added comments on this in the revised manuscript as follows:

“It is interesting to observe fluctuations of *p*TALC and *p*TALD concentrations. This phenomenon seems to be due to the interconversion between *p*TALC and *p*TALD caused by the combined activities of recombinant XMO and endogenous aldehyde-reducing enzymes¹⁷.”

(v) Thank you for pointing this out. This is because the time zero concentration of *p*TALD is the actually measured value (118 mg/L), while we added 200 mg/L. This decrease was due to the time delay in measuring the concentrations in carrying out multiple reactions, while the conversion reaction of *p*TALD to *p*TALC is quite fast. But the key point here is to observe the corresponding reaction through the formation of the corresponding products. Also, there was always certain loss of *p*TALD by evaporation as it is volatile during the handling of the

samples prior to HPLC analysis. To make this point clear, we added explanation in the legend of Supplementary Figure 3 as follows:

“...in the reaction mixtures containing substrates *pX*, *pTALD*, *pTA*, 4-CBAL, *pX* and *pTA*, respectively. The concentrations of substrates added in the reaction mixture are specified in Methods section, while the substrate concentrations shown in above figures are actually measured values. It should be noted in (b) that the initial substrate (*pTALD*) concentration (118 mg/L) is much lower than that (200 mg/L) added in the reaction mixture. This is due to the time delay in measuring the concentrations in carrying out multiple reactions, while the conversion reaction of *pTALD* to *pTALC* is quite fast.”

6. L.147-L.180: though I appreciate that the authors tried several gene/promotor arrangements in their expression plasmids, this part needs to be reduced to at least 50% (better even less). The text is not much more as what can be seen in Fig. 2b/Fig. legends.

[Response]: As we responded earlier, we think that the current level of details is needed for the readers to understand the strategies employed for strain development. If editor thinks that we should reduce the description, we will do so.

7. L.182-187: it is trivial that TA is present as dianion at the pH of culture conditions: reduce this statement to a minimum.

[Response]: Thank you for this comment. Yes we agree that it is obvious to the experts like you. However, to some readers who are not familiar, TPA and TPA disalt might be confusing. The concentrations presented in grams per liter (in toxicity test) can be quite misleading if we do not distinguish TPA vs. TPA disalt clearly.

8. L.196-L.243: Again this part reporting on the optimization of the fermentation should be shortened to 50%, the authors should rather report on the best conditions and then summarize what parameters were essential.

[Response]: As we responded earlier, we think that the current level of details is needed for the readers to understand the strategies employed for bioconversion process. If editor thinks that we should reduce the description, we will do so.

Reviewer #2 (Remarks to the Author):

Our society is totally dependent in a multitude of ways upon chemicals provided by the chemical industry. However, efforts to achieve sustainable development necessitate departure from polluting and high level energy-consuming processes based on non-renewable feedstocks, through exploitation of environmentally friendly chemistry options. A main plank in sustainable chemistry procedures is biocatalysis. Exploitation of biocatalytic procedures requires that they be economically competitive. Significant hurdles for biocatalytic processes for bulk chemicals are modest yields and susceptibility of the biocatalyst to substrate-intermediate-product toxicity.

In this paper, the authors propose a (ultimately) biocatalytic route from the renewable substrate glucose to terephthalic acid (TPA), the monomer of the commodity polymer PET, though they describe here an intermediate process of conversion of p-xylene to TPA, as a proof-of-principle.

The results presented constitute an elegant biocatalysis process based on a rationally designed metabolic pathway constructed in *E. coli* from the upper xylene degradation pathway of *Pseudomonas putida* F1 and the lower p-toluene sulfonate pathway of *Comamonas testosteroni* T-2. The biocatalytic activity of the construct was assessed in a 2-phase bioconversion system in which the xylene was supplied in the apolar phase. Initial experiments revealed bottlenecks within the designed pathway and several inducible constructs were made to relieve them and optimise product yield. The potential toxicity of TPA caused by high substrate feeding rates was analysed and shown to be unproblematic. In one case, a yield of 97% was recorded, which is extraordinary and, if reproducible when upscaled, will represent a breakthrough in hydrocarbon bioconversions.

All in all, this is an excellent and important paper documenting the feasibility of bioconversions for bulk chemistry. While not yet documenting a process of conversion of a renewable substrate to TPA, it shows proof-of-principle, since other publications show the potential feasibility of conversion of renewables to p-xylene. In any case, the exploitation of petroleum constituents as feedstocks for chemicals production, despite being unsustainable over long time spans, is, unlike the use of fossil fuels for energy generation, not a sustainability priority, so a scale-up of the process described in this submission, and its implementation as a viable process for large scale production of TPA, may be envisaged.

[Response]: Thank you so very much for appreciating the importance of our work.

I congratulate the authors on this important advance and only have one, rather trivial question: they examined the toxicity of TPA applied externally but, in this process, TPA is produced internally. Perhaps they may wish to comment on this, and on TPA release after formation?

[Response]: Thank you for the great comment. Indeed, we examined the TPA toxicity to *E. coli* cells by applying TPA desalt externally. As the reviewer knows, it is rather difficult to measure *in vivo* chemical concentration that starts inhibiting cell growth (*in vivo* toxicity). However, *in vitro* toxicity tests, as done for the toxicity measurement to many chemicals by our group and others, at least guide us the approximate tolerance level, which is what we wanted to determine before strain engineering.

Reviewer #3 (Remarks to the Author):

In this manuscript, the authors describe the development of an *E. coli* strain that can convert p-xylene (*pX*) to terephthalic acid (TPA). The engineered pathway relies on known degradative routes for p-xylene and p-toluene sulfonate. The authors first demonstrate independent validation of the up- and downstream pathways, then demonstrate a two-phase partitioning bioreactor to achieve conversion of *pX* to TPA. This is an interesting, largely well-written manuscript. The suitability for this journal arises from the nature of the product being targeted – a very high-volume chemical compound – and the potential for achieving completely bio-based production of TPA. The manuscript would be considerably strengthened by addressing the comments below to improve clarity:

[Response] Thank you very much for your positive comments.

1. p. 4, lines 73 and 74 – I don't think it's quite correct to start that there were "detailed studies" on enzymes in this paper. The pathway relies on known pathways and largely confirms previous observations rather than providing any new biochemical insights. The authors are better served by limiting this statement to "pathway reactions" since that is consistent with both the context and the results.

[Response]: Thank you for the comment. We agree. Thus, the sentence is now changed to "Here we report development of a metabolically engineered *E. coli* strain capable of oxidizing *pX* into TPA, based on detailed studies on pathway reactions ...".

2. p. 5, lines 102-104 – As written, this sentence suggests that BADH does not use NADH and thus would result in a cofactor imbalance. According to Metacyc, the cofactor

requirements for BADH are the same as for the second oxidative step of XMO. Thus, I don't see what using XMO provides an advantage with respect to redox balance.

[Response]: As you can see from several databases including MetaCyc as well as in the literature, XMO converts NADH to NAD⁺ as a cofactor whereas BADH converts NAD⁺ to NADH during the corresponding reactions they catalyze. Thus there is a difference in the cofactor usage between these two enzymes. Thus, the use of XMO in our study is advantageous with respect to redox balance as described in the manuscript.

3. p. 6, lines 121-122 and Supp Fig 3a – Is the production of pTA in this figure actually significant? It seems that a small amount may have been detected at 90 min, but without error bars, it's impossible to know if this is “real”? Were there biological replicates that consistently showed pTA production? Any showing accumulation, i.e., detection at more than one time point?

[Response]: Thank you for the great comment. The purpose of the experiments in Supplementary Fig. 3 was to characterize the successful heterologous expression of enzymes of interest and validation of the expected reactions through detecting the products. We thus focused mainly on detecting the formation of target products during the whole-cell activity assay. In the case of Supplementary Fig. 3a, the products of interest are *pTALC* and *pTALD* as can be seen in our pathway design. It was found that these two metabolites were successfully produced at relatively high yields. So, our purpose was achieved. It was also interesting to find that *pTA* seemed to be accumulated in tiny amount. Thus, we revised our manuscript accordingly. Thank you again.

4. p. 9, line 183 – suggest changing “has not been known” to “has not been evaluated” or something similar.

[Response]: Thank you. Corrected as suggested.

5. p. 9, lines 187-194 and Supp Fig 6 – The data do not support the conclusion that there is no effect on growth at TPA-salt concentrations up to 10 g/L. There is a clear difference in max OD between the 0 g/L control and the 5 and 10 g/L samples. It may be true that the growth rates during exponential phase are not different, but there are not enough data points to accurately calculate a growth rate given that there is only one data point between inoculation at time t=0 and stationary phase. If the authors want to make claims about growth rate, more

data points are needed to ensure that measurements are being made during the exponential phase.

[Response]: Thank you for the great comment. Yes, indeed we agree that we should not use the term “growth rate” with a few sampling points. Thus, as you thankfully pointed out, we changed the description as follows:

“Exposing cells to various concentrations of TPA disodium salts up to 10 g/L (equivalent to TPA concentration of 7.9 g/L) showed only slight reduction in the final optical density. *E. coli* cells could tolerate up to 40 g/L of TPA disodium salts (equivalent to TPA concentration of 31.6 g/L). At this TPA disodium salts concentration, the final optical density was a half that of the control without TPA disodium salts (Supplementary Fig. 6).”

6. p. 15, line 324 – The amount of substrates added need to be provided with more detail than “an appropriate amount.” For each experiment, the specific substrate and amount added should be provide. A similar comment is provided below in reference to Supp Fig 3.

[Response]: Thank you for the comment. As you suggested, we provided the concentrations of substrates in the revised manuscript.

7. p. 17, lines 371-373 – For these latter runs, was there also just a single bolus addition of 50-ml nutrient solution? It is noteworthy that Fig 3 shows no accumulation of glucose after the onset of feeding while Supp Fig 7 clears shows a spike. This suggests a different in either feeding strategy or physiology that should be explained.

[Response]: The latter runs were based on the DO-stat feeding strategy, and thus there was no single bolus addition of nutrient solution. In the case of Fig. 3, the DO-stat nutrient feeding operated perfectly so that the residual glucose was maintained at relatively low concentration. In the case of Supplementary Fig. 7, it belongs to the preliminary runs where the strategy of manual feeding by single bolus addition was applied for nutrient supplementation. Therefore, they are two different modes of nutrient feeding. These were clearly described in our original manuscript.

8. Supp Fig 2 – Additional descriptions are required for the arrows, i.e., specify the proteins that are being pointed to, along with the expected molecular weights.

[Response]: Thank you. As the reviewer suggested, we added the information in the revised manuscript.

9. Supp Fig 3 – More details are needed here about the experimental setup, in particular with respect to the substrates added in each panel, both the identity and amount. Panels 3a are 3e show productivity originating from pX, but pX titer is not shown anywhere. I understand that the solubility is quite low and thus it may have been unmeasurable, but if this is the case, it should be noted in the figure legend and/or the methods. Panel 3a also shows spikes in metabolites, which suggests some sort of feeding regimen. It would also be very helpful to use the same abbreviations (in parentheses) in the figure legend as are used in the main text.

[Response]: Thank you for the comment. Indeed, the reason why *pX* consumption profile was not shown was that the whole-cell activity assay was conducted in aqueous solution in which *pX* could not be dissolved due to its insolubility in water. Thus, it is rather difficult to accurately monitor *pX* consumption as the reviewer noted already. We thus described this in the revised manuscript as the reviewer suggested.

Regarding the spikes of metabolites in Supplementary Fig. 3a, we already responded to the Reviewer 1's comment.

Regarding abbreviations, we corrected our mistakes. Thank you.

10. Supp Table 5 is incorrectly listed as Supp Figure 5. Also, the units for pX added (superscript note 'b') need to be provided.

[Response]: Thank you for pointing out the mistake. We corrected it. Regarding the units for *pX*, we already provided it ("g" for gram) in our original manuscript.

Reviewers' Comments:

Reviewer #1 (Remarks to the Author):

The authors made many improvement and most points have been clarified. They are obviously reluctant to shorten the manuscript, which I still believe would improve it with regard to the general readership of Nature Communication; also no MS-analysis of proteins expressed have been shown.

Surely, there is little doubt that the bands shown in the gel correspond to the respective enzymes. However for a high-quality paper MS-identification is usually standard.

Reviewer #3 (Remarks to the Author):

The authors have done a nice job of revising this manuscript to address the reviewer's comments (particularly Reviewers 1 and 3). There are only two points where there are still lingering questions:

1. p. 5, lines 102-104 – My original comment was as follows: “As written, this sentence suggests that BADH does not use NADH and thus would result in a cofactor imbalance. According to Metacyc, the cofactor requirements for BADH are the same as for the second oxidative step of XMO. Thus, I don't see that using XMO provides an advantage with respect to redox balance.” The authors replied that XMO converts NADH to NAD⁺ while BADH converts NAD⁺ to NADH. If this is the case, these enzymes cannot be catalyzing the same reaction – one would have to be a reduction and the other an oxidation in order to maintain the electron balance – thus, this sentence makes even less sense.

2. p. 6, lines 123-124 and Supp Fig 3a – I understand fully that the point of these experiments was to validate expected reactions; however, the appearance of pTA seems to be an unexpected result. Given that (i) there were no replicates provided, (ii) the absolute amount of pTA detected is so small that it cannot even be quantified from the graph, and (iii) pTA was detected only at one intermediate time point, I recommend softening the language here to say that results suggested the production of pTA but that this observation was not reconfirmed.

Responses to the Comments

Editor's Comments

Dear Dr Lee,

Your manuscript entitled "Biotransformation of *p*-xylene into terephthalic acid by engineered *Escherichia coli*" has now been seen again by our referees, whose comments appear below. In light of their advice I am delighted to say that we are happy, in principle, to publish a suitably revised version in Nature Communications under the open access CC BY license (Creative Commons Attribution v4.0 International License). We therefore invite you to revise your paper one last time to address the remaining concerns of our reviewers. I should emphasize that we may choose to contact our reviewers again before making our final decision. At the same time we ask that you edit your manuscript to comply with our format requirements and to maximise the accessibility and therefore the impact of your work.

[Response]: Thank you very much. We fully revised our manuscript based on the remaining minor points.

As you suggested in the main text and supplementary information, we revised our manuscript based on all the comments. We also confirmed that our manuscript meets format requirements. All the changes made in the manuscript are tracked. The changes in the supplementary information are not tracked as this file will be published as is. The changes made in the supplementary information include: rearranging the supplementary figures, tables and references in the correct order.

As you suggested for adding additional display items, we moved Supplementary Figure 6 into the main text to make it as Fig 3; we also moved Supplementary Table 5 into the main text to make it as Table 1 in the revised manuscript.

Your paper will be accompanied by a two-sentence editor's summary, of between 250-300 characters, when it is published on our homepage. Could you please approve the draft summary below or provide us with a suitably edited version?

[Response]: Thank you very much. You wrote great summary. Only italics were changed as shown in red.

Terephthalic acid (TPA) is an important commodity chemical typically produced from the oxidation of fossil fuel-derived *p*-xylene (*p*X) at high temperature and pressure. Here the

authors report an engineered *Escherichia coli* strain that can transform *pX* into TPA with a high conversion yield.

Reviewers' comments:

Reviewer #1 (Remarks to the Author):

The authors made many improvement and most points have been clarified. They are obviously reluctant to shorten the manuscript, which I still believe would improve it with regard to the general readership of Nature Communication; also no MS-analysis of proteins expressed have been shown. Surely, there is little doubt that the bands shown in the gel correspond to the respective enzymes. However for a high-quality paper MS-identification is usually standard.

[Response]: Thank you for the comment. Indeed, we still think the current level of details is in fact needed for the other researchers to follow what we did for strain development and bioconversion process development. Regarding the MS-analysis of the protein bands, you are right in that providing MS-identification results will be of course much more concrete. However, we already provided suitable evidences through showing (1) proper controls, (2) expected molecular weight bands, and (3) importantly the expected activities together with those of negative controls. We want to thank the reviewer for the great comments that improved our manuscript during our first revision.

Reviewer #3 (Remarks to the Author):

The authors have done a nice job of revising this manuscript to address the reviewer's comments (particularly Reviewers 1 and 3). There are only two points where there are still lingering questions:

1. p. 5, lines 102-104 – My original comment was as follows: “As written, this sentence suggests that BADH does not use NADH and thus would result in a cofactor imbalance. According to Metacyc, the cofactor requirements for BADH are the same as for the second oxidative step of XMO. Thus, I don't see that using XMO provides an advantage with respect to redox balance.” The authors replied that XMO converts NADH to NAD⁺ while BADH converts NAD⁺ to NADH. If this is the case, these enzymes cannot be catalyzing the same reaction – one would have to be a reduction and the other an oxidation in order to maintain the electron balance – thus, this sentence makes even less sense.

[Response]: Thank you for the comment. Sorry for not having been able to properly explain at the first time. Conversions of *pX* to *pTALC* and also *pTALC* to *pTALD* are both oxidation as you know. XMO uses NADH as a cofactor for this oxidation by coupling with oxygen. On the other hand, if BADH was used, obviously cofactor needed for converting *pTALC* to *pTALD* is NAD⁺. Therefore, XMO was employed for two consecutive oxidation steps, which makes the overall pathway redox neutral. We believe that it is now clear. To avoid confusion by the readers, we added “(together with molecular oxygen)” in the revised manuscript page 5 when describing XMO reaction. And we also added “note that BADH would use NAD⁺ for the oxidation reaction converting *pTALC* to *pTALD*.” following the sentence.

2. p. 6, lines 123-124 and Supp Fig 3a – I understand fully that the point of these experiments was to validate expected reactions; however, the appearance of *pTA* seems to be an unexpected result. Given that (i) there were no replicates provided, (ii) the absolute amount of *pTA* detected is so small that it cannot even be quantified from the graph, and (iii) *pTA* was detected only at one intermediate time point, I recommend softening the language here to say that results suggested the production of *pTA* but that this observation was not reconfirmed.

[Response]: Thank you very much for the comment. As the reviewer suggested, we rephrased the sentence in the revised manuscript to “Also, the result suggested production of a small amount of *pTA*, but this observation was not reconfirmed. ”